# Assessment of *Schistosoma mansoni* and soil-transmitted helminth infections and the diagnostic performance of the circulating cathodic antigen test among schoolchildren in Tulla district, Sidama region, Southern Ethiopia

Fitsum Getachew [1,2*], Demissie Assegu[1], Biniyam Kijineh [3], Bamlaku Tadege [1*]

1 School of Medical Laboratory Sciences, Hawassa University, Hawassa, Ethiopia, 2 School of Medical Laboratory Sciences, Wolaita Sodo University, Wolaita, Ethiopia, 3 School of Medical Laboratory Sciences, Wachemo University, Hossana, Ethiopia

* bamlaku22tadege@gmail.com (BT); fitsumgetachew6@gmail.com (FG)

## Abstract

### Background

STH and *S. mansoni* pose significant public health challenges in regions with inadequate sanitation. Existing research on *S. mansoni* intensity remains limited in the study area. *S. mansoni* diagnosis traditionally relies on the KK method, though the POC-CCA urine test offers a rapid alternative with high sensitivity. This study aimed to assess the prevalence and risk factors of *S. mansoni* and STH infections. It also evaluated the intensity of *S. mansoni* and compared the performance of the POC-CCA test to that of KK.

### Methods

A cross-sectional study was conducted among schoolchildren in the Tulla district, Sidama region, from April to June 2024, using a purposive sampling approach. Data were collected through pre-structured questionnaires, and stool samples were analyzed using the KK method, while urine samples were analyzed with the POC-CCA technique. A logistic regression model was employed to examine potential associations between infections and risk factors, the Kappa statistic assessed agreement between tests, and the McNemar chi-square test compared the sensitivity and specificity of the diagnostic methods.

### Result

The prevalence of *S. mansoni* with two combined tests was 36.5% (20.1% by KK and 34.4% by POC-CCA). Activities like irrigation, swimming, and bathing showed significant associations with *S. mansoni* infection. The prevalence of STH was 48.8%,

**Data availability statement:** All relevant data are within the paper and its Supporting information files.

**Funding:** The author(s) received no specific funding for this work.

**Competing interests:** There is no any competing interest for this research. All authors dead and agreed to submit this article to PLoS ONE journal. We read also the journal guideline.

**Abbreviation:** AOR, adjusted odds ratio; CCA, circulating cathodic antigen; CI, Confidence interval; COR, Crude odds ratio; EPG, Eggs per gram; KK, Kato Katz; LR+, Positive likelihood ratio; LR-, N likelihood ratio; MDA, Mass drug administration; NPV, Negative predictive value; NTD, neglected tropical disease; POC, Point of care; PPV, Positive predictive value; SOP, Standard operating procedure; STH, Soil-transmitted Helminth; WHO, World Health Organization; WASH, Water Sanitation and Hygiene; PC, preventive chemotherapy

with *A. lumbricoides* (34%) as the most common, followed by *T. trichuria* (8.2%), hookworm (5.7%), *Taenia* species (1.3%), *H. nana* (0.6%), and *E.vermicularis* (0.2%). Factors like hand washing, fingernail trimming, lack of latrines, and educational status were significantly linked to STH infections. The POC-CCA test demonstrated higher sensitivity (89.6%) than the KK technique (McNemar test $\chi^2$m = 52.3, p < 0.001).

## Conclusions

The moderate prevalence of *S. mansoni* and STH infections, coupled with socio-demographic factors, behaviours, hygiene practices, and sanitation issues associated with these infections, highlights the need for additional control measures beyond deworming. Implementing a highly sensitive POC- CCA test alongside the KK method in low-endemic areas could improve diagnostic accuracy and enhance disease management outcomes.

## Introduction

*Schistosoma* and soil-transmitted helminth (STH) are worm infections that belong to the group of neglected tropical diseases (NTDs) [1]. *Schistosoma mansoni* (*S. mansoni*) is a water-based, vector-borne parasite transmitted indirectly through freshwater snails [2]. People living in areas with inadequate water and sanitation systems are particularly susceptible to this most common and deadly infection [3]. Globally, schistosomiasis is recognized as endemic in 78 low- and middle-income countries in tropical and subtropical areas, estimated to affect more than 251 million people [4]. Furthermore, 243 million people require preventative chemotherapy, including 111 million school-age children, of which 226 million are in Africa [5]. It causes significant morbidity, with an estimated 1.4–3.3 million disability-adjusted life years (DALYs) lost annually due to its health impacts [6].

STHs are intestinal parasitic worms transmitted to humans through fecally contaminated soil [7]. The main STHs that affect humans are *Ascaris lumbricoides* (A. *lumbricoides*), *Trichuris trichuria* (T. *trichuria), Necator americanus, and Ancylostoma duodenale* [8]. These parasites cause gastrointestinal infections and are most commonly found in tropical and subtropical regions, particularly impacting impoverished nations where schoolchildren are especially vulnerable [9].

Globally, an estimated 1.5 billion people are infected with STHs, including 654 million school-aged children living in areas where these parasites are prevalent [10]. The Burden of *Schistosoma* and STHs leads to various health issues, including gastrointestinal bleeding, nutrient malabsorption, loss of appetite, anemia due to the depletion of iron and other essential proteins, growth retardation, impaired cognitive development, school absenteeism, and disability-adjusted life-years lost [11–13].

The World Health Organization (WHO) recommends integrated strategies for controlling and eliminating schistosomiasis and STH. These strategies include preventive chemotherapy (PC), which consists of periodic administration of anthelmintic medicines praziquantel for schistosomiasis and albendazole or mebendazole for STH.

Additionally, improvements in water, sanitation, and hygiene (WASH), behavior change, snail control, and environmental management are essential, as deworming alone cannot prevent re-infection [14,15].

In Ethiopia, an estimated 79 million people live in areas endemic for STHs, and 37.3 million live in regions affected by *S. mansoni*, with the majority being school-aged children—specifically, 25.3 million and 12.3 million children, respectively [16]. National mapping surveys conducted between 2013 and 2015 across Ethiopia's regional states revealed a prevalence of 21.7% for STHs and 3.5% for *S. mansoni* infections among school-aged children [17]. In 2015, Ethiopia launched a national control program aimed at eliminating *S. mansoni* and STHs as serious public health concerns by 2020, with the goal of breaking transmission by 2025 [18]. The program aims to treat at least 75% of school-aged children and to extend treatment to adolescents and adults. It also seeks to integrate initiatives for neglected tropical diseases and strengthen collaboration with mass drug administration efforts. Furthermore, it aims to enhance cooperation with WASH programs [16]. However, the WASH program has limited coverage, particularly in rural areas. This issue is primarily attributed to insufficient collaboration and the high financial costs associated with running the program in Ethiopia.

Results from systematic reviews in Ethiopia have shown that helminth control interventions have significantly reduced the prevalence of STH and *S. mansoni* infections over the years in Ethiopia [3,18]. Additionally, a survey conducted among schoolchildren in our study area revealed a notable decline in the prevalence of STHs (52.4%) and *S. mansoni* (31%) in 2015 compared to the rates in 2007, which were 67.3% for STHs and 76.5% for *S. mansoni* [19,20]. There is annual deworming programme for school children in our study area to control STH and *S. mansoni.* However, the current status of infection intensity in the study area, following the increase in helminth control interventions, remains unknown.

Intestinal *Schistosoma* is diagnosed by microscopic detection of parasite eggs in stool using the Kato-Katz (KK) technique, which is widely used in epidemiological surveys [21]. However, its diagnosis is challenging in low-endemic areas where prevalence and worm burden are reduced, due to factors like uneven egg distribution in stool samples, daily variations in egg excretion, and random distribution effects [22,23]. The point-of-care (POC) cassette test is a relatively new diagnostic tool that detects circulating cathodic antigens (CCA) in urine samples, offering higher sensitivity and operational benefits [23,24]. A past study found that POC-CCA was more sensitive (86% vs. 62%) but less specific (72% vs. ~ 100%) compared to multiple KK smears from a single stool sample. A multivariable modelling study suggested that POC-CCA was significantly more sensitive than KK for detecting low infection intensities (<100 eggs/gram of stool) [25]. A previous study recommended assessing the POC-CCA assay before fully utilizing it for monitoring and evaluating control programs [26].

Therefore, this study aimed to assess the prevalence and risk factors of *S. mansoni* and STH infections among schoolchildren. It also aimed to assess the intensity of *S. mansoni* and compare the diagnostic performance of the POC-CCA test with KK.

## Materials and methods

### Study design and setting

A school-based cross-sectional study was conducted from April to June 2024 among children in four governmental elementary schools in the Tulla district. The study was conducted in four elementary schools in the Tulla district, Sidama region in Southern Ethiopia. The district is situated along the shore of Lake Hawassa and the Tikur Wuha River, approximately 289 kilometers south of Addis Ababa, the capital city of Ethiopia. The altitude is 1800 meters above sea level, with an annual rainfall of 1123 mm and temperatures ranging from 13 to 27°C. Lake Hawassa is the primary water source for the residents of Hawella Tulla, catering to their domestic, agricultural, fishing, and other needs. According to ECSA, the district has a total population of 129,507, comprising 65,018 males and 64,499 females. The district has various social service institutions, including 27 health facilities and 69 schools. The health facilities consist of 1 primary hospital, 6 health centers, 18 rural health posts, 2 urban health posts, and 1 non-governmental health center. In terms of education, the

district includes 21 primary schools, 5 secondary schools, 6 non-governmental primary schools, and 37 non-governmental kindergarten schools (Fig 1).

## Population and enrollment criteria

The source population was all schoolchildren in the Tulla district, Sidama region, southern Ethiopia. The study population included schoolchildren who were enrolled in the selected schools during the study period and met the established inclusion criteria (who had both parental or guardian informed consent and their assent to participate) and exclusion criteria

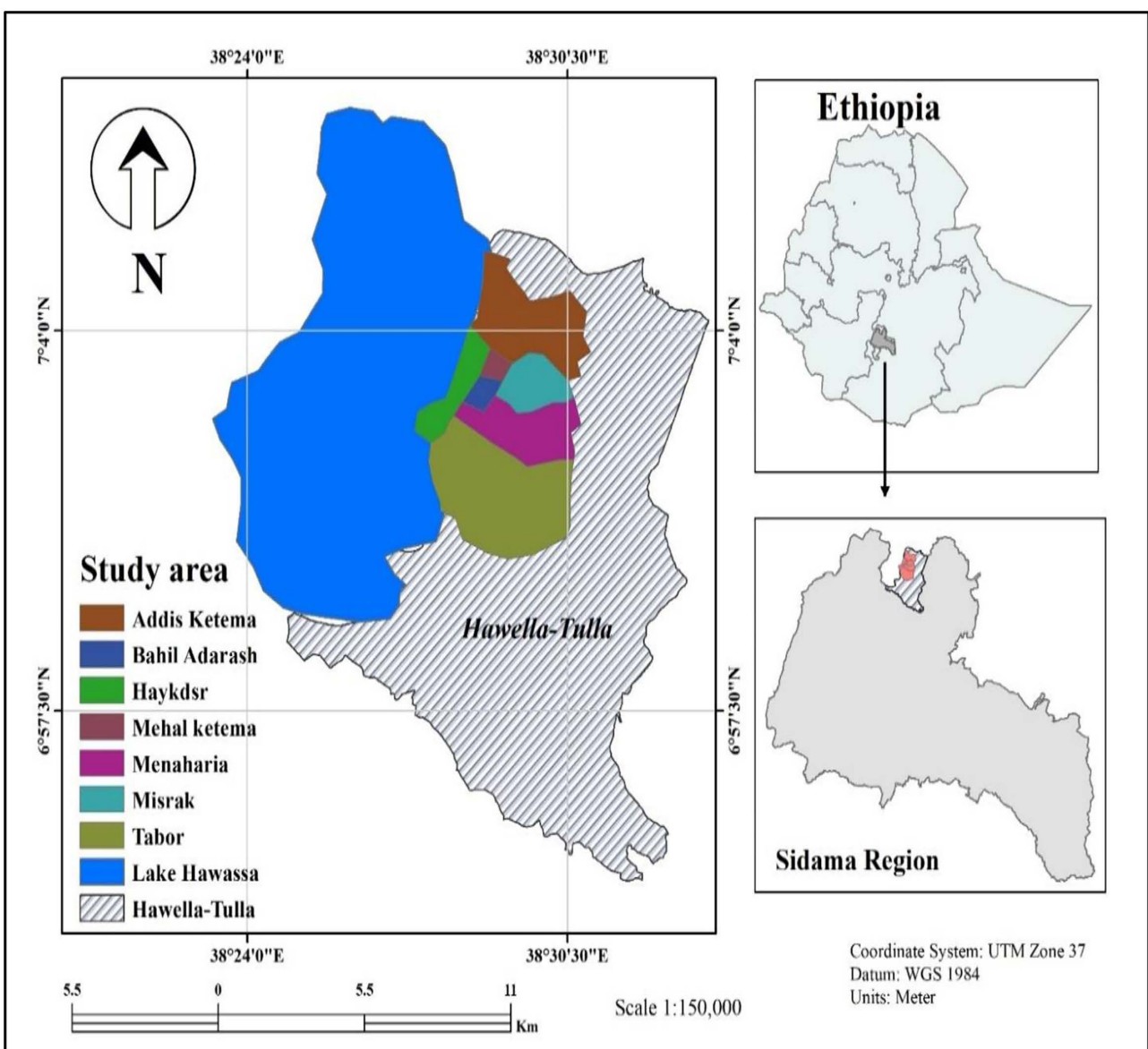

**Fig 1. Map of the study site.** The figure in the left corner depicts Lake Hawassa and the study district, while the upper right corner shows Ethiopia. The figure at the bottom right displays the map of the Sidama region 2024 (The map is prepared using ArcGIS 10.8 (ESRI, Redlands, CA USA) after obtaining country shape files from the humdata.org website (http://data.humdata.org/).

(who had received anti-helminthic treatment within one month before the recruitment period, or who were unable to provide stool and urine samples were excluded from the study).

## Sample size and sampling technique

To determine the sample size, a single population proportion formula was used based on the following assumptions: a prevalence of *S. mansoni* of 31% [20], with a 5% margin of error and a 95% confidence level. As a result, the calculated sample size was 329. Considering the design or cluster effect between schools, the determined number was multiplied by 1.5, and then 5% was added for the non-response rate, giving the final sample size of 484.

To recruit participants for this study, we utilized a purposive sampling technique. In the initial stage, Bushulo and Finchawa, located near the coasts of Lake Hawassa in the Tulla sub-city, were selected alongside two primary schools farther from the lake, Gemeto and Tulla, with student populations of 1,820, 686, 2,050, and 1,439, respectively. Then, the predetermined sample size of 484 was proportionally allocated to the schools, resulting in 147 participants from Bushulo, 55 from Finchawa, 116 from Gemeto, and 166 from Tulla. The students' registration lists served as the sampling frame, and a systematic random sampling technique was employed to select the study subjects from each school. The k-value (sampling interval) was determined by dividing the total number of study participants in each school by the number of students to be selected. The students were then systematically selected at regular intervals (every 12th student), stratified by age (5–19 years) and grade levels (1–4 and 5–8) until the predetermined sample size was achieved. In cases where the selected student was absent from school, three repeated visits were made. When a student missed after three visits, he/she considered non-respondents.

## Dependent and independent variables

The dependent variables were prevalence, infection intensity, and diagnostic performance of *S. mansoni* infection. The independent variables included socio-demographic characteristics such as gender, age, and educational status of both parents, as well as family occupation. Additionally, grade level and the schools attended were taken into account. The distance from home to the lake and practice-related factors were also examined, emphasizing habits such as swimming, fishing, wearing shoes, washing clothes, hand washing before meals and after using the latrine, open-field defecation, trimming fingernails, bathing, and involvement in irrigation activities. Environmental factors considered encompassed the availability of a latrine at home. Clinical symptoms of the disease were also evaluated among study participants.

## Data collection methods

**Interview.** The data was collected using a structured questionnaire. Two trained nurses conducted home-to-home visits to interview the parents or guardians of the children, utilizing the KOBO Collect digital tool. A senior expert and the principal investigator supervised the data collection process, and regular meetings were held to address any challenges and ensure adherence to established protocols and standards.

**Stool and urine sample collection.** Fecal and urine samples were collected in a separate screw-capped, dry, and clean container pre-labelled with the participant's unique identification number. The stool samples were transported within an hour of collection in suitable cool boxes at temperatures between 4°C and 6°C for subsequent examination. Samples from Gemeto and Tulla Elementary Schools were transported to Tulla Hospital, while samples from Bushulo and Finchawa Elementary Schools were sent to the Bushulo Mother and Child Health Specialty Center. The investigator supervised the collection and transportation of samples to ensure compliance with standard operating procedures.

**Stool sample processing using the Kato-Katz technique:** The stool samples were examined using the KK technique to detect parasite eggs. A portion of the sample was processed using a calibrated template holding 41.7 mg of stool. Double slides were prepared for each stool sample. The number of eggs was counted from the double KK microscopic slides, and the average count was converted to eggs per gram of feces (EPG) using a correction factor of 24. Based on WHO cut-off values, the infection intensity was categorized as light (EPG 1–99), moderate (EPG 100–399), or heavy (EPG > 400) [27].

**Urine sample processing using POC-CCA rapid test:** Each urine sample was processed immediately at the schools using the POC-CCA test, following the manufacturer's instructions [28]. This on-site processing enabled prompt results and minimized potential delays or degradation of the samples. 100 µL of urine was transferred to the circular well of the test cassette and allowed to absorb completely into the specimen pad. The test results were read after 20 minutes of assay development. In cases of discrepancies, the tests were re-examined to ensure accuracy. The POC-CCA tests were scored as negative or positive based on the formation and intensity of the control band. Tests were considered invalid if an internal control band did not appear or if results were not read after 25 minutes or more. In such cases, samples were re-run with a new cassette and scored as necessary.

## Data quality assurance

The questionnaire was initially developed in English and then translated into Amharic and Sidamu Afoo, the local language of the participants. It was then translated back into English to ensure consistency. A pre-test was conducted on 5% of the sample size at Tullo Primary School, which was not part of the selected schools for the final study. The questionnaire was subsequently modified based on the findings from the pre-test. The completeness of data collection was overseen daily by the assigned supervisor.

The senior laboratory technologist provided refresher training to laboratory professionals. Standard operating procedures (SOPs) for pre-analytical, analytical, and post-analytical procedures were implemented throughout sample analysis. To ensure internal quality control, 10% of the slides were randomly selected and re-examined by a senior laboratory technologist at Tulla Hospital and Bushulo Mother and Child Health Specialty Centre. Additionally, another experienced technologist at HUCSH conducted an external laboratory evaluation. Both stool and urine samples were collected daily between 09:00–12:00 AM and 03:00–05:00 AM local time throughout the survey.

## Data analysis

Data were entered into the KOBO Collect digital tool and analyzed using IBM SPSS Statistics Version 26. Descriptive statistics, including frequency, percentage, mean, and standard deviation were calculated. Arithmetic mean infection intensity of *S. mansoni* was also amalyzed and reported as low, moderate, or heavy according to WHO guidelines [4]. The mean egg count was reported as a geometric mean. A logistic regression model was used to identify potential determinants of *S. mansoni* and STH infections. Initially, a bivariate analysis was conducted to calculate the crude odds ratios (COR) for each variable and to determine which variables were significantly associated with the outcome. To control for confounding variables, factors with a p-value of ≤ 0.25 from the bivariate analysis were included in the multivariable logistic regression analysis, allowing for the calculation of adjusted odds ratios (AOR). A p-value of less than 0.05 was considered statistically significant. The diagnostic performance of the POC-CCA test was assessed using metrics like sensitivity and specificity, often using the KK as a reference standard [29]. The KK method is the most commonly used diagnostic approach for helminths, epidemiology studies, and treatment evaluation which shows good performance, particularly at detecting helminth infections of moderate and heavy intensity.

Kappa statistic was employed to quantify the agreement between the two diagnostic methods, and the concordance rate was determined as the proportion of cases that yielded identical results. While the McNemar chi-square test was employed to compare the statistically significant differences in sensitivity and specificity between the two diagnostic methods, focusing on cases where the tests disagree.

## Ethical consideration

Ethical clearance was obtained from Hawassa University, College of Medicine, and the other Health Sciences Internal Review Board (IRB) with a reference number: IRB/172/16 in 2024. A letter of support from Hawassa University School of Medical Laboratory was provided to the Sidama Regional State Health Bureau, as well as to the health

and education offices in Tulla district and the directors of the selected schools. Participation was fully voluntary, with written informed consent obtained from the parents/guardians of the children enrolled. An additional separate written assent was secured from children older than 12 years. Explicit written assent was not obtained for children under 12 years, as they may lack the developmental capacity to understand this concept. However, these children were verbally informed about the study in simple language, and their willingness or unwillingness to participate was respected.

Schoolchildren who tested positive for any pathogenic intestinal parasites received appropriate treatment administered by qualified public health professionals. All data collected during the study were kept strictly confidential and used exclusively for research purposes. The study procedures were conducted in full compliance with relevant ethical guidelines and regulatory standards.

## Result

### Socio-demographic characteristics of the study participants

A total of 484 students were invited to participate in the study, of whom 477 met the inclusion criteria and agreed to take part, resulting in a response rate of 98.5%. The study sample comprised 208 males (43.6%) and 269 females (56.4%). Participants' ages ranged from 5 to 19 years, with a mean age of 11.94 years and a standard deviation of 2.76 years. Most of the study participants (62.7%) were 10–14 years old. The majority of students reside within 1 km of a lake (66.5%) and are under the care of their private (daily laborer) parents or guardians (41.7%). The students were selected from 4 primary schools: Bushulo 143 (30%), Finchawa 53 (11.1%), Gemeto 116 (24.3%), and Tulla 165 (34.6%) (Table 1).

**Prevalence of *Schistosoma mansoni* infection.** The prevalence of *S. mansoni* among school children, as determined by the two combined diagnostic methods, was found to be 36.5%. Specifically, 20.1% (96 out of 477 samples) were detected using the KK technique, while 34.4% (164 out of 477 samples) were identified through the POC-CCA test. Compared to sex for *S. mansoni*, boys were more infected than girls (25.0% *vs*. 16.4%). Children aged 10–14 years were more infected (22.4%) than those aged 5–9 years (9.1%). There was a significant variation in the prevalence of *S. mansoni* across different schools (Table 1). The highest prevalence of *S. mansoni* infection was observed among students attending Bushulo Elementary School (36.4%), followed by Finchawa (24.5%), Gemeto (14.6%), and Tulla (8.5%). Children living within 1 km of the lake had a higher *S. mansoni* infection rate (39.6%) compared to those residing more than 1 km away, who had an infection rate of 14.8%. Additionally, children who had greater contact with the contaminated lake exhibited a higher prevalence of *S. mansoni* infection compared to those who had no exposure. Specifically, the infection rate among children who engaged in fishing was 35.1%, notably higher than the 18.1% infection rate among those who did not participate in fishing activities. Table 1 shows detailed information about various activities at the lake and their associated infection rates.

**Potential associated risk factors of *S. mansoni* infections.** In the bivariate logistic regression analysis, several activities were identified as potential risk factors for *S. mansoni* infection. These included washing clothes (13%, 62/477), bathing (31.7%, 151/477), swimming (32.9%, 157/477), fishing (11.9%, 57/477), and engaging in irrigation activities (24.9%, 119/ 477), all with a p-value of ≤ 0.25. Additionally, socio-demographic factors such as gender, age, the parents' educational status, attending Bushulo School and the distance of the residential area from the lake were also identified as potential predictors of *S. mansoni* infection (Table 1).

Multivariate logistic regression analysis was applied to avoid confounding factors and only students engaged in irrigation were 2.4 times more likely to be infected than those not involved (AOR = 2.43; 95% CI: 1.38–4.29, P = 0.002), schoolchildren with swimming habits exhibited a 2.4-fold increase in the likelihood of infection (AOR = 2.41; 95% CI: 1.43–4.34, P = 0.001) and participants who bathed in the lake were 1.8 times more likely to be infected than those who did not engage in this activity (AOR = 1.80; 95% CI: 1.05–3.12, P = 0.033). A school-based analysis revealed that students attending

**Table 1. The distribution of *Schistosoma mansoni* infection by socio-demographic and other factors among schoolchildren in Tulla district, Sidama region, Southern Ethiopia 2024 (n = 477).**

| Variables | S. mansoni | | | COR (95% CI) | p-value | AOR (95%CI) | p-value |
|---|---|---|---|---|---|---|---|
| | Positive % | Negative % | Total % | | | | |
| **Gender** | | | | | | | |
| Male | 52(25.0) | 156(75.0) | 208(43.6) | 1 | | 1 | |
| Female | 44(16.4) | 225(83.6) | 269(56.4) | 0.587(0.374-0.920) | 0.020 | 0.607(0.359-1.027) | 0.063 |
| **Age** | | | | | | | |
| 5-9 | 7(9.1) | 70(90.9) | 77(16.1) | 1 | | | |
| 10-14 | 67(22.4) | 232(77.6) | 299(62.7) | 2.888(1.268-6.578) | 0.012 | 2.118(0.841-5.330) | 0.111 |
| 15-19 | 22(21.8) | 79(78.2) | 101(21.2) | 2.785(1.122-6.914) | 0.027 | 1.442(0.495-4.198) | 0.502 |
| **Name of School** | | | | | | | |
| Bushulo | 52(36.4) | 91(63.6) | 143(30) | 6.163(3.234-11.746) | 0.001 | 4.685(1.869-11.742) | **0.001**** |
| Finchawa | 13(24.5) | 40(75.5) | 53(11.1) | 3.505(1.526-8.051) | 0.003 | 1.444(0.486-4.297) | 0.509 |
| Gemeto | 17(14.6) | 99(85.4) | 116(24.3) | 1.852(0.874-3.926) | 0.108 | 1.528(0.662-3.525) | 0.320 |
| Tulla | 14(8.5) | 151(91.5) | 165(34.6) | 1 | | | |
| **Educational status of the father** | | | | | | | |
| No formal education | 17(18.5) | 75(81.5) | 92(19.3) | 1 | | 1 | |
| Primary school | 58(26.0) | 165(74.0) | 223(46.75) | 1.551(0.846-2.841) | 0.156 | 1.420(0.690-2.921) | 0.341 |
| **Educational status of the mother** | | | | | | | |
| No formal education | 3(7.0) | 40(93.0) | 43(9) | 1 | | 1 | |
| Primary school | 62(23.2) | 205(76.8) | 267(56) | 4.033(1.206-13.485) | 0.024 | 3.391(0.889-12.937) | 0.074 |
| Secondary school | 25(18.8) | 108(81.2) | 133(28) | 3.086(0.883-10.787) | 0.078 | 2.178(0.535-8.859) | 0.277 |
| College and above | 6(17.7) | 28(82.3) | 34(7) | 2.857(0.658-12.397) | 0.161 | 2.384(0.414-13.719) | 0.331 |
| **Distance from residence area** | | | | | | | |
| Within 1 km | 49(30.6) | 111(69.4) | 160(33.5) | 1 | | 1 | |
| More than 1 km | 47(14.8) | 270(85.2) | 317(66.5) | 0.394(0.250-0.623) | 0.001 | 1.194(0.615-2.318) | 0.601 |
| **Bathing in the lake** | | | | | | | |
| Yes | 43(28.5) | 108(71.5) | 151(31.7) | 2.051(1.295-3.248) | 0.002 | 1.804(1.048-3.107) | **0.033*** |
| No | 53(16.3) | 273(83.7) | 326(68.3) | 1 | | 1 | |
| **Swimming habit** | | | | | | | |
| Yes | 50(31.9) | 107(68,1) | 157(32.9) | 2.783(1.780-4.403) | 0.001 | 2.407(1.430-4.052) | **0.001**** |
| No | 46(14.4) | 274(85.6) | 320(67.1) | 1 | | 1 | |
| **Fishing** | | | | | | | |
| Yes | 20(35.1) | 37(64.9) | 57(12) | 2.447(1.345-4.449) | 0.003 | 1.135(0.533-2.417) | 0.743 |
| No | 76(18.1) | 344(71.9) | 420(88) | 1 | | 1 | |
| **Washing clothes along the lake shore** | | | | | | | |
| Yes | 19(30.6) | 43(69.4) | 62(13) | 1.940(1.071-3.513) | 0.029 | 2.016(0.934-4.350) | 0.074 |
| No | 77(17.3) | 338(82.7) | 415(87) | 1 | | 1 | |
| **Involvement in irrigation** | | | | | | | |
| Yes | 44(37.0) | 75(63.0) | 119(25) | 3.452(2.148-5.547) | 0.001 | 2.429(1.377-4.287) | **0.002*** |
| No | 52(14.5) | 306(85.5) | 358(75) | 1 | | 1 | |

AOR, Adjusted Odds Ratio; CI, Confidence interval; COR, Crude Odds Ratio; *, Significant at $p < 0.05$.

Bushulo primary school were 4.7 times more likely to contract *S. mansoni* infection compared to their peers at Tulla primary school (AOR = 4.68; 95% CI: 1.87–11.74, P = 0.001) (Table 1).

### Infection intensity of *Schistosoma mansoni* infection

Out of 174 children infected with *S. mansoni*, 96 tested positive by the KK technique using a standard volume of stool that enables the quantification of ova per gram of stool. The minimum and maximum eggs per gram of stool were 24 and 672, respectively. Likewise, the arithmetic and geometric mean values were 189.75 and 139.39, respectively. According to the WHO classification, among 96 students infected with *S. mansoni*, 33 (34.4%) had light intensity, 54 (56.2%) had moderate intensity, and 9 (9.4%) had heavy intensity (Fig 2).

There was no significant variation in the intensity of infection across age groups. However, the intensity of infection across age groups varies, with the younger age groups, 10–14 years, having less moderate intensity (49.25%) compared to older age groups, 15–19 years (72.7%) and the younger age group, 5–9 years (71.4%). The highest light intensity of *S. mansoni* infection (64.3%) and the lowest moderate intensity (35.7%) were observed among students at Tulla Elementary School, with no cases of heavy intensity when compared to students at other elementary schools (Table 2).

### Prevalence of soil transmitted helminthic infections

Aside from *S. mansoni*, the prevalence of intestinal helminths among the schoolchildren was 233 (48.8%). The most common STHs were *A. lumbricoides* 162 (34%), followed by *T. trichuria* 39 (8.2%) and hookworm 27 (5.7%). There were also intestinal helminth parasites reported in this study including *Taenia* species 6 (1.3%), *Hymenolepis nana* (*H.nana*) 3 (0.6%), and *Enterobius vermicularis* (*E. vermicularis*) 1 (0.2%) (Fig 3). The younger age group of 5–9 years was more infected (49.4%) than those aged 15–19 years (43.6%). The data revealed that children whose parents had no formal

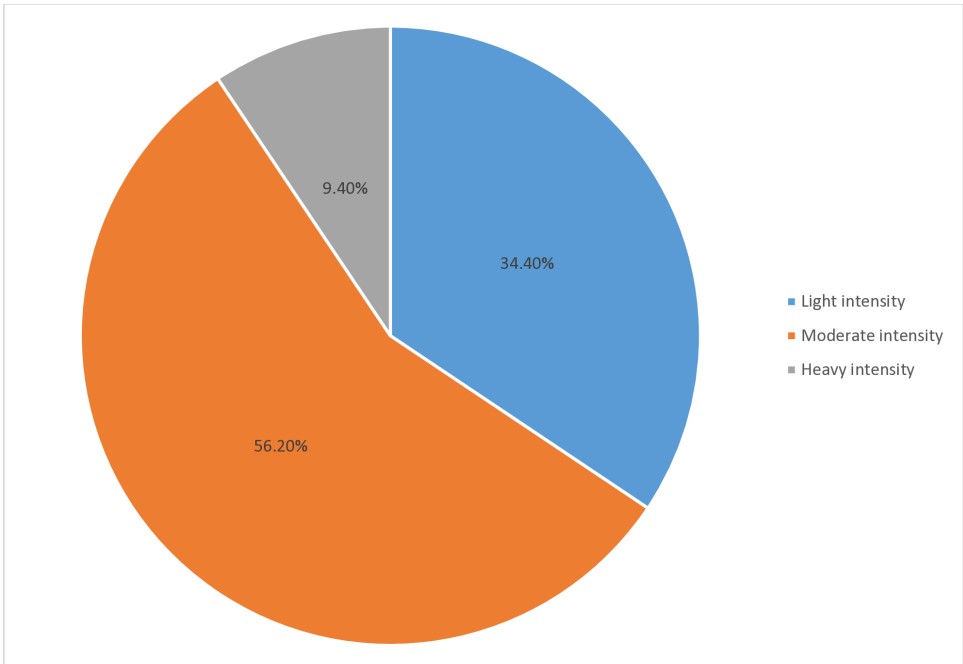

**Fig 2. The prevalence of S. mansoni infection intensity across 477 schoolchildren in Tulla district, Sidama Region, southern Ethiopia, 2024.**

**Table 2. Gender, age, and school-based variation in S. mansoni infection intensity among schoolchildren in Tulla District Sidama Region southern Ethiopia 2024 (n = 477).**

| Variable | N | Intensity of infection n (%) | | |
|---|---|---|---|---|
| | | Light | Moderate | Heavy |
| **Gender** | | | | |
| Male | 52 | 18(34.6) | 28(53.9) | 6(11.5) |
| Female | 44 | 15(34.1) | 26(59.1) | 3(6.8) |
| **Age** | | | | |
| 5-9 | 7 | 2(28.6) | 5(71.4) | 0(0) |
| 10-14 | 67 | 27(40.3) | 33(49.25) | 7(10.45) |
| 15-19 | 22 | 4(18.2) | 16(72.7) | 2(9.1) |
| **Schoolchildren** | | | | |
| Bushulo | 52 | 14(27) | 32(61.5) | 6(11.5) |
| Finchawa | 13 | 4(31) | 7(53.8) | 2(15.2) |
| Gemeto | 17 | 6(35.3) | 10(58.8) | 1(5.9) |
| Tulla | 14 | 9(64.3) | 5(35.7) | 0(0) |

Key – N: refers to the total number of infected; whereas, n: refers to the number of infected according to intensity classes.

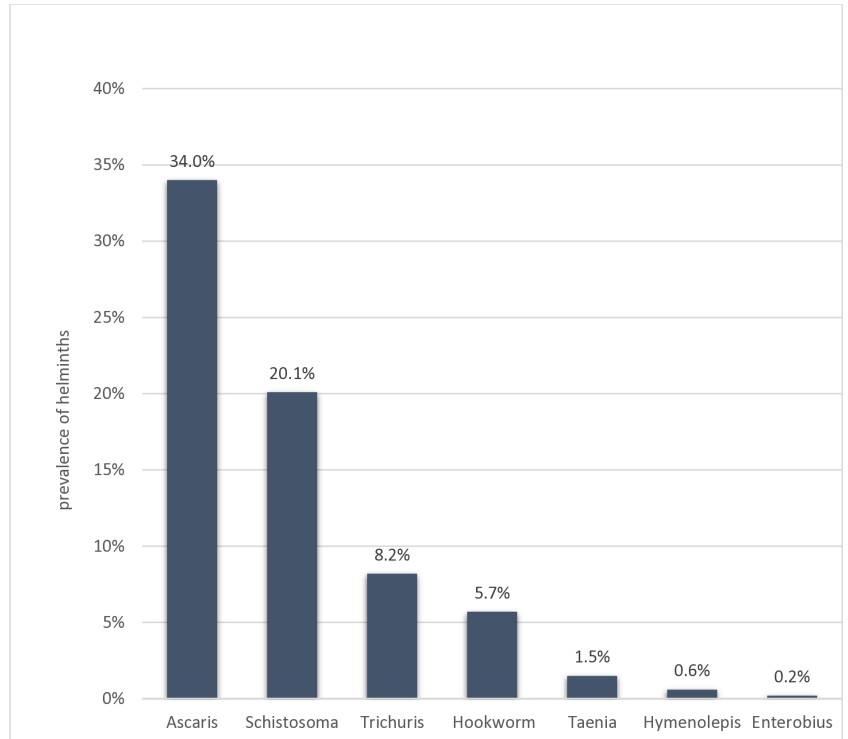

**Fig 3. The prevalence of intestinal helminthic infections across 477 school children in Tulla district, Sidama Region, southern Ethiopia, 2024.**

education were more likely to be infected with STH (59.4%) compared to those whose parents had a college education (27.3%). Additionally, higher STH infection rates were found among children from families where the parents worked as daily labourers (54.5%) or farmers (50.8%). Children who did not wash their hands after using the latrine exhibited a significantly higher STH infection rate of 52.8%, while those who practised hand washing had a much lower rate of 12.8%. Similarly, children who neglected to wash their hands before meals had an infection rate of 53.4%, compared to 33.6% for those who did wash their hands before eating. Furthermore, children who did not trim their nails had an increased risk of contracting STH infections, with an infection rate of 54.7% compared to 32.7% for those who regularly trimmed their nails, as shown in (Table 3).

### Potential associated risk factors of STH infections

In the bivariate logistic regression analysis, both hygiene-related practices and socio-demographic factors were identified as potential risk factors for soil-transmitted helminthic infections. STH infection was associated with latrine use, hand washing practices, fingernail trimming habits, educational level, and occupation, with a p-value of ≤ 0.25 (Table 3).

Finally, the multivariate regression analysis revealed that latrine use, school location, educational status, nail trimming practices, and hand washing after using the latrine were significantly associated with STH infection, with a p-value of ≤ 0.05 as summarised in Table 3. Students from Tula Elementary School had 0.47 times lower odds of STH infection compared to their counterparts at Bushulo Elementary School (AOR = 0.469; 95% CI: 0.256–0.861, P = 0.015). Specifically, children whose fathers had no formal education were found to be 4.2 times more likely to be infected than those whose fathers had completed college or higher education (Adjusted Odds Ratio = 4.161; 95% Confidence Interval: 1.276–13.570, P = 0.018).

The presence of latrines at home also significantly influenced infection rates, with children having access to latrines exhibiting 0.63 times lower odds of infection than those without such facilities (AOR = 0.626; 95% CI: 0.393–0.996, P = 0.048). Furthermore, hygiene practices emerged as critical determinants, as children who washed their hands after using the latrine had a remarkable 0.2 times lower odds of contracting STH infections compared to those who did not (AOR = 0.199; 95% CI: 0.072–0.551, P = 0.002). Similarly, regular nail trimming was associated with a significant reduction in infection risk, with children who trimmed their nails having 0.4 times lower odds of STH infection than those who did not (AOR = 0.413; 95% CI: 0.250–0.682, P = 0.001) (Table 3).

### Detection and performance evaluation of the POC-CCA test

Out of 477 samples examined, the POC-CCA rapid test identified 86 (18.0%) true positives, 303 (63.5%) true negatives, and 10 (2.1%) discrepancies (false negatives) compared to the KK technique. Using KK as the gold standard or 'reference test' [29], the POC-CCA sensitivity was 89.6% (95% CI: 82.3–94.4), specificity 79.5% (95% CI: 75.2–83.3), NPV 96.8% (95% CI: 94.1–98.2), PPV 52.4% (95% CI: 44.6–60.1), Negative LR 0.13 (95% CI: 0.07–0.23) and Positive LR 4.3 (95% CI: 3.38–5.63) (Table 4)

The agreement between the two diagnostic methods was quantified using Cohen's Kappa statistics which yielded a kappa value of 0.55. Furthermore, the overall concordance rate between the KK and POC-CCA was 81.5%. However, to assess the statistical significance of the differences in results between the two tests, we employed the McNemar Chi-square test. The result revealed a value of 52.3.

## Discussion

Although large-scale deworming programmes targeting specific groups have been successful in reducing morbidity associated with *S. mansoni* and STH infections, the absence of a sensitive diagnostic method for detecting low infection intensities and reinfection without additional intervention measures remains unavoidable. This study assessed the prevalence and related risk factors of these parasites, and evaluated the intensity and diagnostic performance of the POC-CCA rapid test for *S. mansoni* among schoolchildren in the Tulla district of the Sidama region, Ethiopia.

**Table 3. Univariate and multivariate logistic regression analyses for socio-demographic and other potential risk factors associated with soil transmitted helminthic infections among school children in Tulla district, Sidama region, Southern Ethiopia 2024 (n = 477).**

| Variables | STH | | | COR (95% CI) | p-value | AOR (95%CI) | p-value |
|---|---|---|---|---|---|---|---|
| | Positive % | Negative % | Total % | | | | |
| Gender | | | | | | | |
| Male | 97(46.6) | 111(53.4) | 208(43.6) | 1 | | | |
| Female | 136(50.6) | 133(49.4) | 269(56.4) | 1.170(0.814-1.681) | 0.395 | | |
| Age | | | | | | | |
| 5-9 | 38(49.4) | 39(50.6) | 77(16.1) | 1 | | | |
| 10-14 | 151(50.5) | 148(49.5) | 299(62.7) | 1.047(0.634-1.728) | 0.857 | | |
| 15-19 | 44(43.6) | 57(56.4) | 101(21.2) | 0.792(0.437-1.437) | 0.443 | | |
| School | | | | | | | |
| Bushulo | 76(53.2) | 67(46.8) | 143(30) | 1 | | 1 | |
| Finchawa | 28(52.8) | 25(47.2) | 53(11.1) | 0.987(0.525-1.857) | 0.969 | | |
| Gemeto | 54(46.5) | 62(53.5) | 116(24.3) | 0.768(0.470-1.254) | 0.292 | | |
| Tulla | 75(45.5) | 90(54.5) | 165(34.6) | 0.735(0.469-1.151) | 0.178 | 0.469(0.256-0.861) | **0.015*** |
| Grade level | | | | | | | |
| 1-4 | 118(48.2) | 127(51.8) | 245(51.4) | 1 | | 1 | |
| 5-8 | 126(54.3) | 106(45.7) | 232(48.6) | 1.279(0.893-1.834) | 0.180 | 0.848(0.495-1.453) | 0.548 |
| Educational status of father | | | | | | | |
| No formal education | 55(59.8) | 37(40.2) | 92(19.3) | 3.964(1.420-11.066) | 0.009 | 4.161(1.276-13.570) | **0.018*** |
| Primary school | 115(51.6) | 108(48.4) | 223(46.7) | 2.840(1.072-7.523) | 0.036 | 2.957(0.991-8.818) | 0.052 |
| Secondary education | 57(40.7) | 83(59.3) | 140(29.4) | 1.831(1.072-7.523) | 0.234 | 1.616(0.547-4.775) | 0.385 |
| College and above | 6(27.3) | 16(72.7) | 22(4.6) | 1 | | 1 | |
| Occupation of family | | | | | | | |
| Civil servant | 6(31.6) | 13(68.4) | 19(4) | 1 | | 1 | |
| Merchant | 56(41.5) | 79(58.5) | 135(28.3) | 1.536(0.550-4.285) | 0.412 | | |
| Farmer | 63(50.8) | 61(49.2) | 124(26) | 2.238(0.799-6.264) | 0.125 | 0.199(0.013-3.028) | 0.245 |
| Private (Daily laborer) | 108(54.3) | 91(45.7) | 199(41.7) | 2.571(0.940-7.037) | 0.066 | 0.262(0.017-3.932) | 0.332 |
| Latrine availability at home | | | | | | | |
| Yes | 128(42) | 177(58) | 305(63.9) | 1 | | 1 | |
| No | 105(61) | 67(39) | 172(36.1) | 0.461(0.315-0.676) | 0.001 | 0.626(0.393-0.996) | **0.048*** |
| Hand washing after latrine | | | | | | | |
| Yes | 6(12.8) | 41(87.2) | 47(9.9) | 1 | | 1 | |
| No | 227(52.8) | 203(47.2) | 430(90.1) | 0.131(0.54-0.315) | 0.001 | 0.199(0.072-0.551) | **0.002*** |
| Shoe wearing | | | | | | | |
| Yes | 198(49.25 | 204(50.75) | 402(84.3) | 1 | | | |
| No | 35(46.7) | 40(53.3) | 75(15.7) | 0.902(0.550-1.477) | 0.681 | | |
| Finger nail trimming | | | | | | | |
| Yes | 39(29.3) | 94(70.7) | 133(27.9) | 1 | | 1 | |
| No | 194(56.4) | 150(43.6) | 344(72.1) | 0.321(0.209-0.493) | 0.001 | 0.413(0.250-0.682) | **0.001*** |
| Hand washing before meal | | | | | | | |
| Yes | 37(33.6) | 73(66.4) | 110(23) | 1 | | 1 | |
| No | 196(53.4) | 171(46.6) | 367(77) | 0.442(0.283-0.890) | 0.001 | 0.585(0.337-1.015) | 0.058 |
| Open field defecation | | | | | | | |
| Yes | 24(48) | 26(52) | 50(10.5) | 1.039(0.578-1.867) | 0.899 | | |
| No | 209(48.9) | 218(51.1) | 427(89.5) | 1 | | | |

Key – AOR: Adjusted Odds Ratio, CI: Confidence interval, COR: Crude Odds Ratio, *: Significant at $p < 0.05$, STH: Soil-transmitted helminths.

**Table 4. Performance of POC-CCA test using Kato-Katz as a gold standard across 477 school children in Tulla district Sidama Region souther Ethiopia 2024.**

| Method | Result | KK (Gold standard) | | Sensitivity % (95% CI) | Specificity % (95% CI) | NPV % (95% CI) | PPV % (95% CI) | LR- % (95% CI) | LR+% (95% CI) |
|---|---|---|---|---|---|---|---|---|---|
| | | Pos | Neg | | | | | | |
| POC-CCA | Pos | 86 | 78 | 89.6 (82.3-94.4) | 79.5 (75.2-83.3) | 96.8 (94.1-98.2) | 52.4 (44.6-60.1) | 0.13 (0.07-0.23) | 4.3 (3.38-5.63) |
| | Neg | 10 | 303 | | | | | | |

Key – KK: Kato-Katz POC-CCA: Point of Care Circulating Cathodic Antigen, NPV: Negative Predictive value, PPV: Positive Predictive value, NLR-: Negative Likely hood ratio, LR+: Positive Likely hood ratio, Pos: Positive, Neg: Negative.

## Prevalence of *Schistosoma mansoni* infection

The prevalence of *S. mansoni* infection was 20.1% (95% CI: 16.8–23.7) using the KK technique. This finding highlights the need for annual deworming of schoolchildren in the study area, as the prevalence falls within the moderate range (≥ 10% but < 50%) according to WHO guidelines [4]. This result is consistent with previous studies conducted in the region, which reported prevalence rates as follows: Jimma. 20.2% [30], Kersa, 19.9% [31], Mekelle, 23.9% [32], Egypt, 19.1% [33], and Angola, 21.2% [34]. However, it is higher than the prevalence observed in Jiga, 15.2% [35], Guangua, 12.6% [36], and Wondo districts, 11.4% [37]. In contrast, the prevalence found in this study is lower than that reported in Lake Tana, 34.9% [38], Jimma town, 28.7% [39], Maksegnit, Debark, Sanja, and Chuahit districts, 33.5% [40], Abbey and Didessa valleys 53.9% [41], and Wolaita, 81.3% [42], Kenya, 53.7% [43], Congo, 90.2% [44], and Tanzania, 90.6% [45]. The variations in the prevalence of *S. mansoni* infection across different regions may be attributed to differences in temperature and rainfall patterns that influence snail reproduction and water availability, which in turn affect the local transmission rates of the parasite. Differences in human behavior towards contaminated water sources include swimming or crossing infested water, as well as performing domestic chores in contaminated water bodies. Inadequate access to safe drinking water and the lack of proper sanitation facilities, leading to open defecation near water sources, influence the transmission cycle of the parasite.

Socioeconomic factors such as increased population density and migration to new localities contribute to the spread of schistosomiasis by increasing the potential for exposure to infested water. Altered agricultural practices and the use of untreated water for irrigation can enhance opportunities for human-water contact and snail proliferation that may also favour the rate of infection. Environmental factors may also impact disease transmission variation across different regions, including the proximity to snail habitats, the construction of dams, land cover (particularly elevation), rainfall patterns, seasonal land surface temperatures, and the presence of flooded agricultural areas. Additionally, it is crucial to consider differences in sample sizes, study designs and diagnostic methods applied.

## Potential associated risk factors of *S. mansoni* infections

A significant association was found between *S. mansoni* infection and schoolchildren who engaged in swimming and bathing activities in the lake, with infection risks being 2.4 times higher for swimming and 1.8 times higher for bathing than those who did not partake in these habits. This association between *S. mansoni* infection and swimming habits among schoolchildren aligns with previous studies [20,36] and is further supported by studies linking bathing habits to infection [38]. Additionally, a strong association was noted between *S. mansoni* infection and participation in irrigation activities, with students involved in irrigation being 2.4 times more likely to be infected than those not engaged in such activities, consistent with prior findings [20,36]. Notable differences in infection rates were observed among students across the various schools in the study areas. Children attending Bushulo Primary School were 4.7 times more likely to be infected with Schistosoma than their peers at Tulla Primary School. This finding aligns with previous studies [20,38]. This might be associated with students' residence distance variation from lakes and other water bodies.

### Intensity of *S. mansoni* infection

In the present study, out of 174 participants infected with *S. mansoni*, 96 were confirmed positive using the KK technique, a standard diagnostic method valued for its ability to quantify EPG of stool. The intensity of infection ranged from 24 to 672 EPG, with an arithmetic mean of 189.75 and a geometric mean of 139.39. The noticeable discrepancy between the arithmetic and geometric means reflects the presence of a right-skewed distribution of egg counts. This pattern is consistent with the typical epidemiology of schistosomiasis, where a minority of individuals harbor high worm burdens, while the majority exhibit low to moderate infection levels [18].

The moderate infection intensity reported in this study is consistent with findings from various regions in Ethiopia [41,46] and in Tanzania [45]. This difference might be due to repeated exposure of schoolchildren to water bodies infested with the infective stage. The intensity of *S. mansoni* infection was found to be slightly higher in male students compared to female students, which is consistent with findings from Kemissie and Wondo Genet [46] and Jimma Town [47] in Ethiopia. The observed difference in proportion might be linked to male students' greater exposure to outdoor activities like fishing, swimming, and irrigation compared to female students.

### Prevalence of soil transmitted helminthic infections

The study area displayed a moderate prevalence of STH infections among schoolchildren despite them being part of the deworming target group. Deworming programs have successfully decreased the global morbidity attributed to STHs [48], but the risk of re-infection persists in the absence of improved water, sanitation, and hygiene WASH practices and health education [49]. A systematic review and meta-analysis estimated that three months after deworming, the prevalence of *A. lumbricoides*, *T. trichuria*, and *hookworms* reached 26%, 36%, and 30% of their pre-treatment levels, respectively. By one-year post-treatment, the prevalence rates approached their pre-treatment levels again, with *A. lumbricoides* at 94%, *T. trichuria* at 82%, and *hookworm* infections at 57% [50]. The occurrence of re-infection highlights the importance of a comprehensive approach to controlling STH. This is especially crucial given the increasing interest in ultimately halting STH transmission in specific geographic regions [51]. The current WHO guidelines recommend the evaluation of any STH infections in school-age children after five to six years of periodic chemotherapy using the standard Kato-Katz thick smear [52].

### Potential associated risk factors of STH infections

The results of this study also provide critical insights into the socio-demographic and environmental determinants associated with STH infections among school children. This research identified several significant factors linked to STH infection, including the lack of a latrine at home, inadequate hand-washing practices following latrine use, the educational status of fathers, improper fingernail trimming practices, and certain school attendance patterns. The lack of a latrine at home was identified as a significant risk factor for STH infections in this study. This finding emphasises the strong association between poor sanitation and the spread of helminths. It agrees with extensive research showing that limited access to sanitation facilities, especially at the household level, significantly contributes to STH infections [53–55]. Without access to latrines, many individuals are forced to practice open defecation, resulting in the direct release of helminth eggs into the environment and promoting continued transmission. Having access to household latrines significantly reduces the risk of exposure to infectious helminth stages in soil and water by minimizing the dispersion of feces in the surrounding area. Notably, This finding aligns with previous research in the same region, which also identified open field defecation as a key factor contributing to STH infection [20]. This study strengthens existing evidence by focusing on sanitation conditions at the household level rather than using broader community-level indicators. This approach provides a deeper understanding of how individual risks are affected by domestic infrastructure, emphasizing the need for targeted public health interventions that go beyond general sanitation promotion. It is essential to prioritize equal access to latrines and ensure their

sustained use at the household level, especially in high-transmission areas where environmental contamination continues to pose a significant challenge.

Hand hygiene practices, especially the failure to wash hands after using the latrine, were significantly associated with the prevalence of STH, which aligns with prior global studies [56–58]. This highlights the importance of hand washing as a key factor in preventing the transmission of STH [59]. In contrast to previous research in the same area, hand hygiene showed no significant effect [20]. This divergence might be due to variations in study design, changes in hygiene behaviors over time, or previous underestimations of individual practices. Furthermore, recent improvements in sanitation infrastructure, in the absence of corresponding behavioral changes, could increase the relative impact of hand hygiene on the risk of infection. These findings indicate the complex interaction between environmental and behavioral factors that influence the transmission of STH and emphasize the importance of ongoing local assessments.

Moreover, the educational status was found to be significantly associated with STH prevalence among children, which aligns with previous studies [60,61]. This highlights the importance of targeting educational programs not only for children but also for parents to foster a supportive environment for health-promoting behaviors. Certain school attendance was also significantly associated with STH infection, in line with previous studies [62,63]. This might be due to differences in sanitation facilities, hygiene practices, socio-economic status, and health education among students, which can significantly influence the transmission dynamics of STH infections in certain schools.

This study also revealed that the practice of not trimming fingernails was significantly associated with STH infection, which aligns with previous studies [54,64–66]. Since long fingernails can harbor soil and pathogens, serving as a reservoir for helminth eggs that can be ingested or transmitted through hand-to-mouth contact, public health initiatives need to promote personal hygiene practices among school-aged children, including regular nail-trimming and cleanliness.

## Comparison of the POC-CCA test performance with KK on *S. mansoni*

The current study found a better diagnostic performance of the POC-CCA assay using a single urine sample than duplicate KK thick smears from a single stool sample. From the 96 KK positive samples, the POC-CCA test identified 10 (10.4%) as negative. Conversely, with KK negative results, POC-CCA classified only 78 (4.9%) as positive, indicating that the KK technique has higher specificity.

In this study, using the KK technique as the gold standard, the POC-CCA test demonstrated higher sensitivity but notably lower specificity. The improved sensitivity (particularly in detecting low-intensity infections) might be attributed to the POC-CCA test's ability to detect trace levels of circulating antigens in urine, allowing for broader detection of *Schistosma* infection, including cases that KK might miss due to the known variability in egg excretion and the limited sensitivity of KK, especially in low-endemic settings [67]. However, the specificity of the POC-CCA test was found to be 79.6%, which is consistent with previously reported ranges but remains relatively low [25]. This reduced specificity may be attributed to cross-reactivity with other helminth infections, the subjective interpretation of trace results as positive, and the continued excretion of CCA following recent treatment or in the absence of active infection [68]. Such limitations can result in false positives, particularly in low-endemic or post-treatment settings, potentially leading to an overestimation of prevalence. Despite its limitations, the POC-CCA test is a valuable diagnostic tool in field settings because of its ease of use, rapid results, and high sensitivity. However, its limited specificity highlights the necessity for confirmatory testing, especially in situations involving surveillance, program evaluation, and low-transmission areas where diagnostic accuracy is crucial.

The overall agreement between KK and POC-CCA was moderate, as reflected by a kappa statistic of 0.55 (p < 0.001) and an observed agreement rate of 81.5%. While this level of concordance suggests a reasonable overlap between the two diagnostic tools, it must be interpreted cautiously given the limitations of κ statistics, which are sensitive to disease prevalence and imbalanced marginal totals [69]. McNemar's test revealed a statistically significant discordance between the two diagnostic methods, with a chi-square value of 52.3 and p < 0.001. This discrepancy was primarily driven by the high number of cases that were positive by the point-of-care circulating cathodic antigen (POC-CCA) test but negative

by the KK method. This lack of alignment underscores the fundamentally different targets of the two diagnostic tools: the POC-CCA test detects active infections by identifying CCA in urine, while the KK method relies on detecting *Schistosoma* eggs in stool, which can vary from day to day and show inconsistencies within the same sample [22]. Therefore, in field settings, especially in low-endemic or post-treatment areas, the POC-CCA test may identify infections that the KK method misses, making it a more sensitive tool for surveillance and monitoring of control programs.

## Conclusions

The existence of moderate prevalence of *S. mansoni* and STH infections among school children emphasises the need of annual deworming for school-aged children in the study area. The findings from this study also provide valuable insights into socio-demographic factors, behavioural factors, personal hygiene practices and environmental sanitation association to STH and Schistosoma infections in these children. This highlights the need for additional control measures to supplement deworming programmes, including health education focused on personal hygiene and environmental sanitation. Furthermore, this study shows that KK has low sensitivity and may underestimate the accurate prevalence. In contrast, the POC-CCA test demonstrates higher sensitivity in the study area, making it a more reliable option for case detection and large-scale screening. Therefore, we recommend adopting the POC-CCA test in school-based *S. mansoni* infection control and elimination programmes alongside the KK method, as this could enhance diagnostic accuracy and improve management outcomes. However, to optimise the consistency and utility of this test, further research to standardise the scoring system is strongly recommended.

## Limitations

This study has a few limitations that should be acknowledged. (i), the cross-sectional design provides a snapshot of *S. mansoni* and STH infection at a single point in time, which may not capture seasonal variations or fluctuations in infection rates over longer periods. (ii), the use of a single stool sample per participant for the KK technique may have led to underestimation of infection prevalence and intensity, given the known day-to-day variability in helminth egg excretion. (iii), the purposive selection of schools limits the generalizability of the findings to the larger community in the study area.

## Supporting information

**S1 Table. The questionnaire on socio-economic, environmental and hygienic practice.**
(DOCX)

**S1 Data. The orginal raw data file of the research.**
(SAV)

## Acknowledgments

Primarily, we would like to thank Hawassa University, School of Medical Laboratory Sciences, for funding this research. Then our thanks go to the study participants and the school community for their cooperation, as their contributions were crucial to its success. Finally, we would also like to thank Tulla Primary Hospital and Bushulo Mother and Child Health Specialty Center staff, especially laboratory professionals, for their dedication and hard work in collecting and conducting experiments.

## Author contributions

**Conceptualization:** Bamlaku Tadege, Fitsum Getachew.

**Data curation:** Bamlaku Tadege, Fitsum Getachew, Biniyam Kijineh.

**Formal analysis:** Bamlaku Tadege, Fitsum Getachew, Biniyam Kijineh.

**Investigation:** Fitsum Getachew.

**Methodology:** Bamlaku Tadege, Fitsum Getachew, Demissie Assegu, Biniyam Kijineh.

**Supervision:** Bamlaku Tadege, Demissie Assegu.

**Visualization:** Fitsum Getachew, Demissie Assegu.

**Writing – original draft:** Bamlaku Tadege, Fitsum Getachew.

**Writing – review & editing:** Bamlaku Tadege, Demissie Assegu, Biniyam Kijineh.

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
