## [Decision Letter · Decision Letter 0]

30 Jun 2025

Dear Dr. Tadege,

Thank you for submitting your manuscript to PLOS ONE. After careful consideration, we feel that it has merit but does not fully meet PLOS ONE’s publication criteria as it currently stands. Therefore, we invite you to submit a revised version of the manuscript that addresses the points raised during the review process.

We look forward to receiving your revised manuscript.

Kind regards,

Abayeneh Girma

Academic Editor

PLOS ONE

Journal Requirements: 

3. We note that your Data Availability Statement is currently as follows: [data are all contained within the manuscript and/or Supporting Information files, enter the following: All relevant data are within the manuscript and its Supporting Information files.]

Additional Editor Comments:

A major revision is required.

Reviewers' comments:

Reviewer's Responses to Questions

**Comments to the Author**

1. Is the manuscript technically sound, and do the data support the conclusions?

Reviewer #1: Yes

Reviewer #2: Yes

2. Has the statistical analysis been performed appropriately and rigorously?

Reviewer #1: Yes

Reviewer #2: Yes

3. Have the authors made all data underlying the findings in their manuscript fully available?

Reviewer #1: Yes

Reviewer #2: No

4. Is the manuscript presented in an intelligible fashion and written in standard English?

Reviewer #1: Yes

Reviewer #2: No

Reviewer #1: The manuscript is well written, the methodology, results, statistical analysis and discussion were well presented. The manuscript is technically sound and data provided in the manuscript. The statistical analysis is appropriate. All findings were outlined in the manuscript. The manuscript is presented in a standard English.

The authors should address the comments made on the manuscript.

Reviewer #2: General comments

The manuscript presents a well-conducted cross-sectional study on the prevalence, intensity, and risk factors of Schistosoma mansoni and soil-transmitted helminth (STH) infections among schoolchildren in Ethiopia, alongside an evaluation of the diagnostic performance of the Point-of-Care Circulating Cathodic Antigen (POC-CCA) test. The study addresses an important public health issue in a region where these infections are endemic. The methodology is robust, and the findings are relevant to ongoing control efforts. However, there are several areas where the manuscript could be improved to enhance clarity, accuracy, and impact.

1. Ethical considerations: The manuscript states that ethical clearance was obtained from Hawassa University, but the reference number provided (IRB/172/16) appears outdated (2016). Given that the study was conducted in 2024, clarification or updated ethical approval documentation is necessary. Again, the consent process is described, but it would be beneficial to explicitly state whether assent was obtained from all children, including those under 12, or only those aged 12 and above.

2. Data availability: The manuscript claims that "all relevant data are within the manuscript and its Supporting Information files," but no supplementary files are provided. The authors should either upload the supporting data or clarify where it can be accessed.

3. Diagnostic performance analysis: The comparison between POC-CCA and Kato-Katz is a key contribution, but the specificity of POC-CCA (79.6%) is notably low. The authors should discuss potential reasons for this (e.g., cross-reactivity, persistent antigens post-treatment) and implications for field use. Again, the McNemar’s test result (χ² = 52.3, p < 0.001) indicates significant disagreement between the two tests, but the discussion does not adequately address this. The authors should elaborate on the practical implications of this discrepancy.

4. Risk factor analysis: The association between STH infections and factors like latrine availability and handwashing is well-documented in the literature. The authors should contextualize their findings more critically, discussing how their results align with or diverge from previous studies in similar settings. Again, the multivariate analysis for S. mansoni risk factors is clear, but the odds ratios for some variables (e.g., swimming, irrigation) are high. The authors should consider potential confounding factors (e.g., frequency of water contact) that were not adjusted for.

5. Prevalence and intensity: The reported prevalence of S. mansoni (20.1% by Kato-Katz) is described as "moderate," but the WHO threshold for moderate prevalence is ≥10% and <50%. The authors should explicitly state how their findings align with WHO guidelines for mass drug administration (MDA). Again, the intensity classification (light, moderate, heavy) is based on WHO criteria, but the geometric mean egg count (139.39 EPG) suggests a skewed distribution. The authors should discuss whether this affects the interpretation of intensity.

6. Limitations: The cross-sectional design and purposive school selection are noted as limitations, but the authors should also address potential biases (e.g., seasonal variations in transmission, non-random sampling). Again, the use of a single stool sample for Kato-Katz may underestimate prevalence due to day-to-day variation in egg excretion. This should be discussed as a limitation.

7. Minor Comments: (i) Abstract: The abstract is comprehensive but could be more concise. The methods section, for example, includes unnecessary details (e.g., SPSS version). Again, the conclusion should avoid redundancy (e.g., "necessity for additional control measures" is repeated). (ii) Tables and Figures: Table 1 (infection intensity) is clear but could be merged with Table 2 (socio-demographic factors) to streamline the presentation. Again, Figure 3 (helminth distribution) is informative but would benefit from a clearer legend (e.g., full parasite names instead of abbreviations). (iii) Writing and Clarity: Some sentences are overly long or awkwardly phrased (e.g., "The absence of a latrine at home emerged as a prominent risk factor for STH infection"). Editing for conciseness would improve readability. Again, the abbreviation "STH" is used throughout but should be defined at first mention in the abstract and main text. (iv) References: References are generally appropriate, but some are outdated (e.g., WHO guidelines from 2011). The authors should cite more recent WHO publications (e.g., 2022 guidelines). Again, a few references are incomplete (e.g., "Weekly epidemiological record Relevé épidémiologique hebdomadaire" lacks volume/issue details).

**Do you want your identity to be public for this peer review?** For information about this choice, including consent withdrawal, please see our Privacy Policy

Reviewer #1: **Yes: ** Obiageli Josephine Nebe

Reviewer #2: No

---

## [Author Response · Author response to Decision Letter 1]

20 Aug 2025

Response to the PLoS ONE editor and reviewers

First and foremost, we sincerely thank you for reviewing our manuscript and offering valuable comments and suggestions. Your insights have been very helpful in highlighting areas for improvement, and we have carefully addressed each of the points you raised.

We followed our manuscript to meet PLOS ONE's style requirements, including those for file naming.

The corresponding author's ORCID ID is addressed

3. We note that your Data Availability Statement is currently as follows: [data are all contained within the manuscript and/or Supporting Information files, enter the following: All relevant data are within the manuscript and its Supporting Information files.]

Please confirm at this time whether or not your submission contains all raw data required to replicate the results of your study. Authors must share the “minimal data set” for their submission. PLOS defines the minimal data set to consist of the data required to replicate all study findings reported in the article, as well as related metadata and methods

We included raw data used for this manuscript results analysis

Reviewers' comments:

Reviewer's Responses to Questions

Comments to the Author

1. Is the manuscript technically sound, and do the data support the conclusions?

Reviewer #1: Yes

Reviewer #2: Yes

We thank all reviewers

2. Has the statistical analysis been performed appropriately and rigorously?

Reviewer #1: Yes

Reviewer #2: Yes

We thank all reviewers

3. Have the authors made all data underlying the findings in their manuscript fully available?

Reviewer #1: Yes

Reviewer #2: No

Thank you for your curiosity. We attached raw data and supplementary information, like the questionnaire

4. Is the manuscript presented in an intelligible fashion and written in Standard English?

Reviewer #1: Yes

Reviewer #2: No

We revised our manuscript once again for any topographical and grammatical corrections in the revised manuscript and indicated on the marked revised manuscript

5. Review Comments to the Author

Reviewer #1: The manuscript is well written, the methodology, results, statistical analysis and discussion were well presented. The manuscript is technically sound, and data provided in the manuscript. The statistical analysis is appropriate. All findings were outlined in the manuscript. The manuscript is presented in Standard English.

The authors should address the comments made on the manuscript.

Reviewer #2: General comments

The manuscript presents a well-conducted cross-sectional study on the prevalence, intensity, and risk factors of Schistosoma mansoni and soil-transmitted helminth (STH) infections among schoolchildren in Ethiopia, alongside an evaluation of the diagnostic performance of the Point-of-Care Circulating Cathodic Antigen (POC-CCA) test. The study addresses an important public health issue in a region where these infections are endemic. The methodology is robust, and the findings are relevant to ongoing control efforts. However, there are several areas where the manuscript could be improved to enhance clarity, accuracy, and impact.

1. Ethical considerations: The manuscript states that ethical clearance was obtained from Hawassa University, but the reference number provided (IRB/172/16) appears outdated (2016). Given that the study was conducted in 2024, clarification or updated ethical approval documentation is necessary. Again, the consent process is described, but it would be beneficial to explicitly state whether assent was obtained from all children, including those under 12, or only those aged 12 and above.

We want to clarify that the ethical clearance reference number is not outdated; it is based on the Ethiopian calendar (E.C.). The reference number: IRB/172/16 E.C. corresponds to the Gregorian calendar date of 22/03/2024, which is indicated on the ethical clearance letter. We add E.C. to indicate the calendar, see the correction on line number 242 in the revised manuscript. Additionally, we acknowledge that our initial manuscript did not mention that assent was obtained from students aged ≥12 years following parent/guardian consent. This has now been addressed and appropriately revised in the manuscript. See the revised one from line number 245 – 247

2. Data availability: The manuscript claims that "all relevant data are within the manuscript and its Supporting Information files," but no supplementary files are provided. The authors should either upload the supporting data or clarify where it can be accessed.

Regarding the comment on the supplementary files, we have separately supplementary information. We also attached raw data

3. Diagnostic performance analysis: The comparison between POC-CCA and Kato-Katz is a key contribution, but the specificity of POC-CCA (79.6%) is notably low. The authors should discuss potential reasons for this (e.g., cross-reactivity, persistent antigens post-treatment) and implications for field use. Again, the McNemar’s test result (χ² = 52.3, p < 0.001) indicates significant disagreement between the two tests, but the discussion does not adequately address this. The authors should elaborate on the practical implications of this discrepancy.

Regarding the specificity of the POC-CCA test (79.6%), which is notably low, we were asked to discuss potential reasons and its implications for field use. We have addressed this from line numbers 497 – 521in the revised manuscript as follows: “However, the specificity of the POC-CCA test was found to be 79.6%, which is consistent with previously reported ranges but remains relatively low (20). This reduced specificity may be attributed to cross-reactivity with other helminth infections, the subjective interpretation of trace results as positive, and the continued excretion of CCA following recent treatment or in the absence of active infection (64). Such limitations can result in false positives, particularly in low-endemic or post-treatment settings, potentially leading to an overestimation of prevalence. Despite this, the POC-CCA test remains a practical diagnostic tool in field settings due to its operational simplicity, rapid results, and high sensitivity. However, its limited specificity underscores the importance of confirmatory testing, especially in surveillance, program evaluation, and low-transmission areas where diagnostic accuracy is critical.” Regarding the McNemar’s test result (χ² = 52.3, p < 0.001), we were informed that the discussion did not adequately address the observed discrepancy or elaborate on its practical implications. Therefore, we have addressed the issue as follows: “Importantly, McNemar’s test revealed a statistically significant discordance between the two methods (χ² = 52.3, p < 0.001), driven primarily by the high number of POC-CCA positive but KK negative cases. This lack of alignment between the two diagnostic tools underscores their fundamentally different targets: POC-CCA detects active infection through CCA in urine, whereas KK relies on the detection of Schistosoma eggs in stool, which are subject to day-to-day variation and intra-sample inconsistencies (17). Therefore, in field settings, particularly in low-endemic or post-treatment areas, the POC-CCA test may identify infections that KK misses, making it a more sensitive tool for surveillance and monitoring of control programs.”

4. Risk factor analysis: The association between STH infections and factors like latrine availability and handwashing is well-documented in the literature. The authors should contextualize their findings more critically, discussing how their results align with or diverge from previous studies in similar settings. Again, the multivariate analysis for S. mansoni risk factors is clear, but the odds ratios for some variables (e.g., swimming, irrigation) are high. The authors should consider potential confounding factors (e.g., frequency of water contact) that were not adjusted for.

As we were asked to critically analyse the association between STH infections and factors like latrine availability and hand washing, and to discuss how our results compare or differ from previous studies in similar settings, we addressed this in lines 445–472 of the revised manuscript as follows:

The lack of a latrine at home was identified as a significant risk factor for soil-transmitted helminth (STH) infections in this study. This finding highlights the strong connection between poor sanitation and the transmission of helminths. It aligns with extensive research showing that limited access to sanitation facilities, especially at the household level, plays a major role in driving STH infections (49–51). Without access to latrines, many individuals are forced to practice open defecation, resulting in the direct release of helminth eggs into the environment and promoting continued transmission. Having access to household latrines significantly reduces the risk of exposure to infectious helminth stages in soil and water by minimizing the dispersion of feces in the surrounding area. Notably, This finding aligns with previous research in the same region, which also identified open field defecation as a key factor contributing to the prevalence of soil-transmitted helminths (STH) (22). This study reinforces existing evidence by examining sanitation status at the household level instead of relying on broader community-level indicators. By doing so, it provides a deeper understanding of how individual risks are influenced by domestic infrastructure. This highlights the need for targeted public health interventions that extend beyond general sanitation promotion. It is essential to prioritize equal access to latrines and ensure their sustained use at the household level, especially in high-transmission areas where environmental contamination continues to pose a significant challenge.

“Hand hygiene practices, especially the failure to wash hands after using the latrine, were significantly associated with the prevalence of STH, which aligns with prior global studies (52–54). This highlights the importance of handwashing as a key factor in preventing the transmission of STH (55). In contrast to previous research in the same area, hand hygiene showed no significant effect (22). This divergence may be due to variations in study design, changes in hygiene behaviors over time, or previous underestimations of individual practices. Furthermore, recent improvements in sanitation infrastructure, in the absence of corresponding behavioral changes, could increase the relative impact of hand hygiene on the risk of infection. These findings underscore the complex interaction between environmental and behavioral factors that influence the transmission of soil-transmitted helminths (STH) and emphasize the importance of ongoing local assessments.

We were advised to consider potential confounding factors, such as frequency of water contact; however, this variable was not included in the final analysis as it was removed after the pretest due to participants’ poor recall.

5. Prevalence and intensity: The reported prevalence of S. mansoni (20.1% by Kato-Katz) is described as "moderate," but the WHO threshold for moderate prevalence is ≥10% and <50%. The authors should explicitly state how their findings align with WHO guidelines for mass drug administration (MDA). Again, the intensity classification (light, moderate, heavy) is based on WHO criteria, but the geometric mean egg count (139.39 EPG) suggests a skewed distribution. The authors should discuss whether this affects the interpretation of intensity.

As we were asked to explicitly state how the 20.1% prevalence of S. mansoni aligns with WHO guidelines for MDA, we note that this has already been addressed in the discussion section of the previously submitted manuscript. We were also asked whether the interpretation of infection intensity was affected by the skewed distribution when using arithmetic and geometric means. we addressed this in lines 401–404 of the revised manuscript as follows: “The noticeable discrepancy between the arithmetic and geometric means reflects the presence of a right-skewed distribution of egg counts. This pattern is consistent with the typical epidemiology of schistosomiasis, where a minority of individuals harbor high worm burdens, while the majority exhibit low to moderate infection levels (42).

6. Limitations: The cross-sectional design and purposive school selection are noted as limitations, but the authors should also address potential biases (e.g., seasonal variations in transmission, non-random sampling). Again, the use of a single stool sample for Kato-Katz may underestimate prevalence due to day-to-day variation in egg excretion. This should be discussed as a limitation.

Regarding the comments on the limitations section of the manuscript, we addressed this in lines 532–535 in the revised manuscript as follows: “This study has a few limitations: (i), the cross-sectional design provides a snapshot of S. mansoni and STH infection at a single point in time, which may not capture seasonal variations or fluctuations in infection rates over longer periods. (ii), the use of a single stool sample per participant for the KK technique may have led to underestimation of infection prevalence and intensity, given the known day-to-day variability in helminth egg excretion. (iii), the purposive selection of schools limits the generalizability of the findings to the larger community in the study area.

7. Minor Comments: (i) Abstract: The abstract is comprehensive but could be more concise. The methods section, for example, includes unnecessary details (e.g., SPSS version). Again, the conclusion should avoid redundancy (e.g., "necessity for additional control measures" is repeated). (ii) Tables and Figures: Table 1 (infec

---

## [Decision Letter · Decision Letter 1]

10 Sep 2025

Dear Dr. Tadege,

Thank you for submitting your manuscript to PLOS ONE. After careful consideration, we feel that it has merit but does not fully meet PLOS ONE’s publication criteria as it currently stands. Therefore, we invite you to submit a revised version of the manuscript that addresses the points raised during the review process.

We look forward to receiving your revised manuscript.

Kind regards,

Abayeneh Girma

Academic Editor

PLOS ONE

Journal Requirements:

Reviewers' comments:

Reviewer's Responses to Questions

**Comments to the Author**

Reviewer #1: All comments have been addressed

Reviewer #3: (No Response)

Reviewer #4: (No Response)

2. Is the manuscript technically sound, and do the data support the conclusions?

Reviewer #1: Yes

Reviewer #3: Yes

Reviewer #4: Partly

3. Has the statistical analysis been performed appropriately and rigorously?

Reviewer #1: Yes

Reviewer #3: No

Reviewer #4: Yes

4. Have the authors made all data underlying the findings in their manuscript fully available?

Reviewer #1: Yes

Reviewer #3: Yes

Reviewer #4: Yes

5. Is the manuscript presented in an intelligible fashion and written in standard English?

Reviewer #1: Yes

Reviewer #3: Yes

Reviewer #4: Yes

Reviewer #1: The authors have adequately addressed the comments raised previously, however there few comments that need to be clarified in the current review. I found the manuscript technically sound and have value for policy and public health practice.

Reviewer #3: I advise the author(s) to take into account the following issues even if they made revisions to the text in response to the feedback and recommendations of earlier reviewers:

1. Uncorrected or Inaccurate Responses to Reviewers:

Ethical approval reference still unclear despite E.C. explanation; conversion not standardly recognized without explicit note.

Data availability statement contradictory: claims data in SI, but no files uploaded; "available upon request" violates PLOS policy.

Raw data not publicly accessible; IRB restriction cited but not justified sufficiently for non-public sharing.

No supplementary files provided despite mention of questionnaire and raw data.

Use of McNemar’s test inappropriate for described analysis; original use of logistic regression more suitable, change unexplained.

2. New Comments/Questions to Improve Quality:

Correct inconsistent verb tense usage (e.g., "data was analysed" → "data were analysed").

Clarify statistical methods: justify switch from logistic regression to McNemar’s test.

Fix subject-verb agreement errors (e.g., "the results shows" → "the results show").

Provide DOI or repository link for raw data to comply with PLOS policy.

Include full ethical approval document with date conversion proof as supplementary.

Define assent process clearly for children under 12 in methods.

Update all WHO references to 2022 guidelines and complete missing citation details.

Reviewer #4: Major Comments

1. What is your main objective? This might result in major revision of the whole MS contents.

a. To assess the prevalence of S. mansoni and STHs (POC-CCA is additional)

b. To evaluate the diagnostic performance of KK vs POC-CCA (STH is additional)

c. ???

2. Your result part is highly fragmented or not well organized: needs reshuffling.

3. What is the relevance of your study

4. Your discussion is precise and not attractive for your readers

Minor comments

Title

1. What do you mean by short title?

2. Check affiliation 3

Abstract

1. A sentence but two different ideas ‘While S. mansoni infection leads to chronic illness and mortality, research on its intensity remains limited in the study area.’

2. Write acronyms of Point of Care Circulating Cathodic Antigen Test (POC-CCA) it its first appearance (in background section).

3. Modify your language through your entire document (e.g. A school-based cross-sectional study was conducted among 477 school children from April to June 2024 in Tulla district, Sidama Region, Southern Ethiopia).

4. Modify: Urine samples were also collected from the study participants and tested using POC-CCA for the detection of Schistosoma antigens.

5. Other helminths included … seams meaningless??

6. Avoid vague sentences: e.g. ‘Moderate S. mansoni and STH infection and poor hygiene practice underscore for additional control measures.

7. Your conclusion is not attractive.

8. Reconsider your key words (read about how to select key words).

9. Geneal comment: Critical language problems

Introduction

Background

1. Put references in [..].

2. Paragraph 5 (line 106-111): It looks like you are talking about your own study

3. The last paragraph: It is the main objective of your study but not well structured, and I recommend re-write. Are you comfortable with ‘to evaluate the prevalence…’? Also check your language (S. mansoni, S. mansoni).

Method

1. … among four governmental elementary school children (schools)

2. Add sample collection portion and clearly indicate how you collected stool and urine samples from your study participants.

3. Infection intensities: You did categories (light, moderate and heavy) only for S. mansoni?

4. What is the relationship between quality and sample collection time?

5. Data analysis: indicate your reference test and give details about your analysis.

Results

1. I prefer if you put your results of diagnostics for S. mansoni and then to risk factors, STH,…

2. Table 1: move it under risk factors sub-title.

3. What is the importance of table 2?

**Do you want your identity to be public for this peer review?** For information about this choice, including consent withdrawal, please see our Privacy Policy

Reviewer #1: **Yes: ** Obiageli Josephine Nebe

Reviewer #3: **Yes: ** Alqeer Aliyo

Reviewer #4: No

---

## [Author Response · Author response to Decision Letter 2]

16 Oct 2025

PONE-D-25-23474R1

Assessment of Schistosoma mansoni and Soil-transmitted Helminth Infections and the Diagnostic Performance of the Circulating Cathodic Antigen Test among Schoolchildren in Tulla District, Sidama Region, Southern Ethiopia. PLOS ONE

Response to the PLoS ONE editor and reviewers' comments and questions

First and foremost, we sincerely thank you for reviewing our manuscript and offering valuable comments and suggestions. Your insights have been very helpful in highlighting areas for improvement, and we have carefully addressed each of the points you raised. We respond to each comment, suggestion and question raised by reviewers, and these responses are highlighted with green colour.

We thank you for your scientific contribution

Journal Requirements:

If the reviewer comments include a recommendation to cite specific previously published works, please review and evaluate these publications to determine whether they are relevant and should be cited.

Recommendation given by reviewer I to state background information if the two organisms are problems in the area under study. Information on ongoing interventions (in terms of number of rounds of treatment, Vector WASH services) on these infections should be mentioned and implications for the prevalence recorded in the present study should be investigated and reported.

We include the current global controlling strategies and background information on the prevalence of STH and Schistosoma in Ethiopia and in the current study area and control intervention measures at the national level and in the region of our study area in the introduction part from line number 118 to 147 which is highlighted in revised marked manuscript and we cite new references used for background information (reference numbers 14 – 20) in the reference section

We cited references which are relevant to our manuscript. All references cited in this manuscript are revised to ensure its completeness and correctness to fit the PLOS ONE reference citation procedure; for this, it is highlighted in the revised marked references section

1. If the authors have adequately addressed your comments raised in a previous round of review and you feel that this manuscript is now acceptable for publication, you may indicate that here to bypass the “Comments to the Author” section, enter your conflict of interest statement in the “Confidential to Editor” section, and submit your "Accept" recommendation.

Reviewer #1: All comments have been addressed

Reviewer #3: (No Response)

Reviewer #4: (No Response)

We revised the manuscript to address the constructive comments and suggestions provided by the reviewers. Thank you for your time and dedication

2. Is the manuscript technically sound, and do the data support the conclusions?

Reviewer #1: Yes

Reviewer #3: Yes

Reviewer #4: Partly

We revised the comments raised by the reviewer concerning this part and conclusion part is revised in advance

3. Has the statistical analysis been performed appropriately and rigorously?

Reviewer #1: Yes

Reviewer #3: No

Reviewer #4: Yes

We thank you all reviewers

In the revised manuscript, we clearly explain why we apply McNemar’s test analysis and its purpose. We also explain the purpose of using logistic regression analysis which is indicated in data analysis section

4. Have the authors made all data underlying the findings in their manuscript fully available?

Reviewer #1: Yes

Reviewer #3: Yes

Reviewer #4: Yes

We thank you all reviewers

5. Is the manuscript presented in an intelligible fashion and written in Standard English?

Reviewer #1: Yes

Reviewer #3: Yes

Reviewer #4: Yes

We thank all reviewers

6. Review Comments to the Author

Reviewer #1: The authors have adequately addressed the comments raised previously; however there few comments that need to be clarified in the current review. I found the manuscript technically sound and have value for policy and public health practice.

Thank you, Reviewer. We include the current global controlling strategies and background information on the prevalence of STH and Schistosoma in Ethiopia and in the current study area and control intervention measures at the national level and in the region of our study area in the introduction part from line number 118 to 147 which is highlighted in revised marked manuscript and we cited new references used for background information (reference numbers 14 – 20) in the reference section. We also added additional keywords significantly related to the study findings. We agreed to keep the research title as it is. ________________________________________

Reviewer #3: I advise the author(s) to take into account the following issues even if they made revisions to the text in response to the feedback and recommendations of earlier reviewers:

1. Uncorrected or Inaccurate Responses to Reviewers:

Ethical approval reference still unclear despite E.C. explanation; conversion not standardly recognized without explicit note.

Thank you, Reviewer, for your time and concern

In our initial submission, we uploaded the ethical clearance approval letter provided by the IRB of Hwassa University College of Medicine and Health Science, which was issued in 2024. So, I missed uploading the ethical clearance approval letter in the second submission. In the current revised manuscript submission, I have uploaded approval letter to the other files. The reference number provided on the ethical clearance application letter is not related to the calendar. Instead, they are unique code number assigned to the ethical approval letter by the IRB. If you require confirmation, you can contact the IRB office at this email address: embialle@hu.ede.et

Data availability statement contradictory: claims data in SI, but no files uploaded; "available upon request" violates PLOS policy.

Raw data not publicly accessible; IRB restriction cited but not justified sufficiently for non-public sharing.

No supplementary files provided despite mention of questionnaire and raw data.

All data used for this manuscript have been submitted. We provided the raw data and questionnaire in a supplementary file during the second manuscript submission and we have no remaining data to submit. We have resubmitted the raw data and questionnaire in the supplementary files in the current revised manuscript submission.

Use of McNemar’s test inappropriate for described analysis; original use of logistic regression more suitable, change unexplained.

We used SPSS regression analysis for descriptive analysis and for the significant association between risk factors and infection, whereas the McNemar chi-square test was employed to compare the statistically significant differences in sensitivity and specificity between the two diagnostic methods, focusing on cases where the tests disagree. This is clearly described in the data analysis section of this revised manuscript.

2. New Comments/Questions to Improve Quality:

Correct inconsistent verb tense usage (e.g., "data was analysed" → "data were analysed").

Subject-verb agreement is corrected in the revised manuscript

Clarify statistical methods: justify the switch from logistic regression to McNemar’s test.

This one is described above; We used SPSS regression analysis for descriptive analysis and for the significant association between risk factors and infection, whereas the McNemar chi-square test was employed to compare the statistically significant differences in sensitivity and specificity between the two diagnostic methods, focusing on cases where the tests disagree. This is clearly described in the data analysis part of this revised manuscript.

Fix subject-verb agreement errors (e.g., "the results shows" → "the results show").

Subject-verb agreement is corrected in the revised manuscript

Provide DOI or repository link for raw data to comply with PLOS policy.

Our college is planning to develop a repository link; for this, we have submitted raw data in a supplementary file in this revised manuscript.

Include full ethical approval document with date conversion proof as supplementary.

This is described above. We uploaded the ethical clearance approval letter provided by the IRB of Hwassa University College of Medicine and Health Science, which was issued in 2024 in this revised manuscript. We address this issue in the ethical part. Ethical clearance was obtained from Hawassa University, College of Medicine, and the other Health Sciences Internal Review Board (IRB) with a reference number: IRB/172/16 in 2024. If you require confirmation, you can contact the IRB office at this email address.

Define the assent process clearly for children under 12 in methods

We explain the assent process clearly for children under 12 in methods as follow:

Explicit written consent was not obtained for children under 12 years, as they may lack the developmental capacity to understand this concept. However, these children were verbally informed about the study in simple language, and their willingness or unwillingness to participate was respected. This is included in the ethical clearance part

Update all WHO references to 2022 guidelines and complete missing citation details.

We revised not only the WHO references but all other references following the Vancouver citation style, following PLOS ONE guidelines, which are highlighted in marked revised manuscript.

WHO updated citation references include reference numbers 1 and 4 in the revised manuscript

Reviewer #4: Major Comments

1. What is your main objective? This might result in major revision of the whole MS contents.

a. To assess the prevalence of S. mansoni and STHs (POC-CCA is additional)

b. To evaluate the diagnostic performance of KK vs POC-CCA (STH is additional)

c. ???

The objectives of this study are:

To assess the prevalence of S. mansoni and STH infections.

To assess risk factors for STH and S. manson infections

To evaluate the intensity of S. mansoni infection

To compare the diagnostic performance of POC-CCA test to that of KK.

We clearly address these objective in the abstract and last paragraph of introduction section

2. Your result part is highly fragmented or not well organised: needs reshuffling.

We revised the results section and reorganized it to make the information clearer and more attractive for the reader.

3. What is the relevance of your study

Our study relevant to current WHO NTD control program as regional and local information are important assess ongoing controlling stratigies for the following reasons:

1. WHO has launched deworming programs for Schistosomiasis and soil-transmitted helminths targeting vulnerable school-age children? Although these deworming efforts have successfully reduced morbidity caused by Schistosoma and STH, re-infection continues in the absence of complementary measures to sustain deworming. This study is important for providing valuable supplementary interventions in the regional study area.

2. Intestinal schistosomiasis and soil-transmitted helminth infections are diagnosed by detecting parasite eggs in stool samples through microscopic methods, primarily using the Kato-Katz (KK) technique. Recommended by the WHO, this method is commonly used in epidemiological surveys. However, the Kato-Katz technique has low sensitivity, making diagnosis difficult in areas with low prevalence and decreased worm loads. This difficulty results from several factors, including uneven egg distribution in stool samples, daily variations in egg excretion, and the random nature of egg dispersal. A more sensitive diagnostic method, called the POC-CCA test, has been developed to detect S. mansoni antigens in urine samples, especially useful in low-endemic areas, either as a replacement or supplement to the Kato-Katz technique. Nonetheless, the diagnostic accuracy of this method has not been thoroughly evaluated across different regions. This study is relevant to address this gap.

4. Your discussion is precise and not attractive to your readers.

We revised the discussion section and provided detailed descriptions of specific subtopics, which are highlighted in the marked revised manuscript. Additionally, we added subsections to improve readability (see from number line 563 – 606 in the marked revised manuscript)

Minor comments

Title

1. What do you mean by short title?

Short title is requested by PLoS ONE submission guidelines, which means writing the research title in a short sentence, keeping the main idea of the original research title

2. Check affiliation 3

We check, and it is correct

Abstract

1. A sentence but two different ideas. While S. mansoni infection leads to chronic illness and mortality, research on its intensity remains limited in the study area.’

We revised it and cleared the sentences in the current revised manuscript in the abstract part

2. Write acronyms of Point of Care Circulating Cathodic Antigen Test (POC-CCA) in its first appearance (in the background section)

We corrected acronyms

3. Modify your language throughout your entire document (e.g. a school-based cross-sectional study was conducted among 477 school children from April to June 2024 in Tulla district, Sidama Region, Southern Ethiopia).

Once again, we revised our manuscript throughout the document and marked the changes in the revised document.

4. Modify: Urine samples were also collected from the study participants and tested using POC-CCA for the detection of Schistosoma antigens.

We changed to urine samples were analyzed with POC-CCA technique.

5. Other helminths included … seams meaningless??

Yes, it may not be clear to all readers, so we removed others in the abstract part and replaced them with the name of the parasite species in the revised manuscript section

6. Avoid vague sentences: e.g. ‘Moderate S. mansoni and STH infection and poor hygiene practice underscore for additional control measures.

Yes, we revised and cleared the sentence in the revised manuscript, which is highlighted in the marked manuscript

7. Your conclusion is not attractive.

We revised and r

---

## [Decision Letter · Decision Letter 2]

16 Nov 2025

Assessment of Schistosoma mansoni and Soil-transmitted Helminth Infections and the Diagnostic Performance of the Circulating Cathodic Antigen Test among Schoolchildren in Tulla District, Sidama Region, Southern Ethiopia.

PONE-D-25-23474R2

Dear Dr. Tadege,

We’re pleased to inform you that your manuscript has been judged scientifically suitable for publication and will be formally accepted for publication once it meets all outstanding technical requirements.

Kind regards,

Abayeneh Girma

Academic Editor

PLOS ONE

Additional Editor Comments (optional):

Reviewers' comments:

Reviewer's Responses to Questions

**Comments to the Author**

Reviewer #1: All comments have been addressed

2. Is the manuscript technically sound, and do the data support the conclusions?

Reviewer #1: Yes

3. Has the statistical analysis been performed appropriately and rigorously?

Reviewer #1: Yes

4. Have the authors made all data underlying the findings in their manuscript fully available?

Reviewer #1: Yes

5. Is the manuscript presented in an intelligible fashion and written in standard English?

Reviewer #1: Yes

Reviewer #1: The manuscript looks great and all comments made in the previous document addressed. Good effort, great work.

**Do you want your identity to be public for this peer review?** For information about this choice, including consent withdrawal, please see our Privacy Policy

Reviewer #1: **Yes: ** Obiageli Josephine Nebe

---

## [Editor Report · Acceptance letter]

PONE-D-25-23474R2

PLOS ONE

Dear Dr. Tadege,

I'm pleased to inform you that your manuscript has been deemed suitable for publication in PLOS ONE. Congratulations! Your manuscript is now being handed over to our production team.

Kind regards,

on behalf of

Dr. Abayeneh Girma

Academic Editor

PLOS ONE